# The Impact of Cerebral Perfusion on Mesenchymal Stem Cells Distribution after Intra-Arterial Transplantation: A Quantitative MR Study

**DOI:** 10.3390/biomedicines10020353

**Published:** 2022-02-01

**Authors:** Ilya L. Gubskiy, Daria D. Namestnikova, Veronica A. Revkova, Elvira A. Cherkashova, Kirill K. Sukhinich, Mikhail M. Beregov, Pavel A. Melnikov, Maxim A. Abakumov, Vladimir P. Chekhonin, Leonid V. Gubsky, Konstantin N. Yarygin

**Affiliations:** 1Department of Neurology, Neurosurgery and Medical Genetics, Department of Medical Nanobiotechnology, Pirogov Russian National Research Medical University of the Ministry of Healthcare of the Russian Federation, 117977 Moscow, Russia; dadnam89@gmail.com (D.D.N.); tcherelvira@gmail.com (E.A.C.); abakumov1988@gmail.com (M.A.A.); chekhoninnew@yandex.ru (V.P.C.); gubskii@mail.ru (L.V.G.); 2Radiology and Clinical Physiology Scientific Research Center, Federal Center of Brain Research and Neurotechnologies of the Federal Medical Biological Agency, 117513 Moscow, Russia; mik.beregov@gmail.com; 3Cell Technology Laboratory, Federal Research and Clinical Center of Specialized Medical Care and Medical Technologies of the Federal Medical Biological Agency of Russian Federation, 115682 Moscow, Russia; veronicarevkova@gmail.com; 4Laboratory of Problems of Regeneration, Koltzov Institute of Developmental Biology of the Russian Academy of Sciences, 119334 Moscow, Russia; sukhinichkirill@gmail.com; 5Department of Fundamental and Applied Neurobiology, Serbsky Federal Medical Research Centre of Psychiatry and Narcology of the Ministry of Healthcare of Russian Federation, 119034 Moscow, Russia; proximopm@gmail.com; 6Drug Delivery Systems Laboratory, D. Mendeleev University of Chemical Technology of Russia, 125047 Moscow, Russia; 7Laboratory of Cell Biology, Orekhovich Institute of Biomedical Chemistry of the Russian Academy of Sciences, 119121 Moscow, Russia; kyarygin@yandex.ru; 8Department of General Pathology and Pathophysiology, Russian Medical Academy of Continuous Professional Education, 125284 Moscow, Russia

**Keywords:** stroke cell therapy, selective intra-arterial perfusion, transcatheter intra-arterial perfusion, MRI, mesenchymal stem cells, intra-arterial, cell transplantation, MCAO

## Abstract

Intra-arterial (IA) mesenchymal stem cells (MSCs) transplantation providing targeted cell delivery to brain tissue is a promising approach to the treatment of neurological disorders, including stroke. Factors determining cell distribution after IA administration have not been fully elucidated. Their decoding may contribute to the improvement of a transplantation technique and facilitate translation of stroke cell therapy into clinical practice. The goal of this work was to quantitatively assess the impact of brain tissue perfusion on the distribution of IA transplanted MSCs in rat brains. We performed a selective MR-perfusion study with bolus IA injection of gadolinium-based contrast agent and subsequent IA transplantation of MSCs in intact rats and rats with experimental stroke and evaluated the correlation between different perfusion parameters and cell distribution estimated by susceptibility weighted imaging (SWI) immediately after cell transplantation. The obtained results revealed a certain correlation between the distribution of IA transplanted MSCs and brain perfusion in both intact rats and rats with experimental stroke with the coefficient of determination up to 30%. It can be concluded that the distribution of MSCs after IA injection can be partially predicted based on cerebral perfusion data, but other factors requiring further investigation also have a significant impact on the fate of transplanted cells.

## 1. Introduction

Transplantation of mesenchymal stem cells (MSCs) is a novel approach to the treatment of many severe neurological disorders causing irreversible damage of neural tissue, including traumatic brain injury, spinal cord injury, multiple sclerosis, neurodegenerative diseases, and stroke [1,2,3]. In the case of stroke, different routes of MSCs administration, including direct intracerebral, intrathecal, intraventricular, intravenous, intra-arterial, intranasal, and others have been tested in animal studies [4,5]. Among them, the intra-arterial (IA) delivery appears one of the most effective, probably because it guarantees targeted cell delivery to the brain vascular system [6,7,8]. Moreover, rapid development of the endovascular mechanical thrombectomy technique and its widespread use in acute stroke treatment made IA transplantation more feasible for routine clinical practice [9,10]. According to the ClinicalTrials.gov site, five-phase I and II clinical trials focused on IA administration of MSCs or bone marrow mononuclear cells into patients with subacute or chronic ischemic stroke have been registered throughout the world and some of them have been already completed (reviewed in [4,11]). The published results of the clinical trials indicate that IA transplantation of cells is essentially safe in humans and can promote some neurological improvement after stroke. However, the obtained results on the efficacy of the treatment were not conclusive [12,13,14,15,16,17]. One of the most likely reasons for the uncertainty of the trial results may be the imperfect study design recommending suboptimal patient recruitment criteria, cell dose, time window, number and frequency of transplantations, and other parameters [4]. Further investigation and better understanding of the mechanisms of MSCs’ therapeutic effects, as well as the assessment of cell distribution and homing after transplantation may help to resolve these issues [5].

The factors determining the distribution of transplanted MSCs within the brain after IA administration have not been fully elucidated. Recently, Walczak et al. [18] demonstrated that cerebral perfusion may be one of the key factors influencing the intra-brain distribution of IA administered stem cells in large and small animal models and suggested its estimation by transcatheter intra-arterial perfusion magnetic resonance imaging (MRI) to predict cell allocation after IA administration. Quite probably, along with brain tissue perfusion, some other factors, e.g., varying adhesion of cells to vessel walls determined by the concentration of certain cell adhesion molecules on the apical surface of endothelium or secretion of chemotactic recruitment factors by injured brain tissue may also affect distribution of IA transplanted cells [19,20]. In this work, we aimed to quantitatively assess the impact of brain tissue perfusion on the distribution of IA transplanted MSCs in intact rats and rats with experimental stroke. To do this, we performed selective MR-perfusion study with bolus injection of gadolinium-based contrast agent and evaluated the correlation between different perfusion parameters and the distribution of MSCs in the brain estimated by susceptibility weighted imaging (SWI) immediately after cell transplantation.

## 2. Materials and Methods

### 2.1. Cell Culture

MSCs were isolated from human placenta as described previously [21]. Cells were placed in culture flasks with the complete culture medium DMEM-F12 supplemented with 2 mM L-glutamine, 100 U/mL penicillin, 0.1 mg/mL streptomycin, and 10% fetal bovine serum (all reagents from Gibco) and maintained in a humidified atmosphere under standard conditions (37 °C, 5% CO_2_). Cells were passaged at 80% confluence and collected for transplantation after 3–5 passages. The phenotype of cultured cells checked by flow cytometry was CD34−, CD45−, HLA-DR−, CD105+, CD29+, CD73+, CD90+, which is typical for MSCs. Prior to transplantation cells were labeled with superparamagnetic iron oxide (SPIO) microparticles (MC03F Bangs Laboratories, mean diameter 0.50 ± 0.99 μm) carrying the Dragon Green fluorescent dye (λex = 480 nm, λem = 520 nm) and with the red lipophilic membrane fluorescent dye PKH26 (Sigma-Aldrich, Burlington, MA, USA) as described previously [22]. Double cell labeling had no influence on cell viability and proliferation [7,22]. Before transplantation, cell viability was checked by the trypan blue test using an automated cell counter (Invitrogen) and was more than 90%. A dose of 5 × 10^5^ cells in 2 mL of saline was prepared for each IA transplantation.

### 2.2. Animals

Male Wistar rats weighing 250–300 g (*n* = 10) were purchased from AlCondi, Ltd., Moscow, Russia. The animals were housed in groups of four to five animals per cage before surgery and individually after the transient middle cerebral occlusion (tMCAO) operation. They were kept under standard conditions-12-h/12-h light/dark cycle, room temperature 22 ± 2 °C, humidity 45–65%, and free access to standard rodent chow and water. Experiments were carried out in accordance with the guidelines of the Declaration of Helsinki and directive 2010/63/EU on the protection of animals used for scientific purposes of the European Parliament and the Council of European Union dated 22 September 2010, and approved by the Pirogov Russian National Research Medical University Animal Care and Use Commission (protocol code No 13/2020 from 8 October 2020). All surgical procedures (stroke modeling and IA transplantation of MSCs) and MRI studies were conducted under inhalation anesthesia (Aerrane, Baxter HealthCare Corporation, Deerfield, IL, USA) supplied by the animal anesthesia system (E-Z-7000 Classic System, E-Z-Anesthesia^®^ Systems, Palmer, PA, USA): 3.5–4% isoflurane mix with atmospheric air for stroke modeling or with pure oxygen for MRI for the induction of anesthesia and 2–2.5% isoflurane/air or oxygen mix for its maintenance. Body temperature was maintained around 37 °C with a heating pad to prevent hypothermia during surgery and MRI. The animal procedures were of medium severity, caused short-term medium-level pain or stress, and all efforts were made to minimize the number of animals in the experiment and exclude painful manipulations and other unpleasant effects. The researchers provided close control over the condition of animals, and, in case signs of pain or distress were detected, an analgesic (meloxicam) was administered to relieve them. At the end of the experiment, rats were euthanized using an induction chamber (E-Z-7000 Classic System, E-Z-Anesthesia^®^ Systems, Palmer, PA, USA) and inhalation anesthesia with a lethal dose of isoflurane. Afterwards, just before the transcardial perfusion animals were additionally injected with the lethal dose of Zoletil. All work involving animals was reported according to the ARRIVE guidelines.

### 2.3. Study Design

Rats (*n* = 10) were randomly divided into two experimental groups: (1) intact rats (*n* = 3) and (2) rats with experimental stroke modeling using transient 90 min tMCAO (*n* = 7, 1 of which was excluded from the experiment due to formation of a subarachnoid hemorrhage after surgery). IA transplantation of MSCs into rats with experimental stroke was performed 24 h after stroke modeling. The time window for transplantation, as well as cell dose of 5 × 10^5^ in 2 mL of saline for all rats were chosen according to the results of our previous studies [7,23]. After insertion of the IA catheter into the external carotid artery, anaesthetized and ready for IA infusion rats were placed into the MRI scanner. The initial part of MR-examination included acquisition of the following MR sequences: isotropic T2-weighted images (T2wi) for estimation of brain anatomy and confirmation of lesion formation in case of stroke modeling, diffusion-weighted images (DWI), and susceptibility weighted images (SWI) for visualizing the original state of the brain before the injection of cells, and, finally, the selective intra-arterial perfusion study with bolus contrast injection. After the bolus contrast injection, 1 mL of saline was slowly manually injected to remove residual contrast from the catheter and 5 min pause was made to let the remnants of the contrast wash out from the brain. IA transplantation of MSCs suspension in 2 mL of saline was performed also inside the MRI scanner over a 20 min time period using the nanoinjector. The infusion parameters were selected based on our previous experiments [7,23] and literature data [6]. After the end of cell injection, the second part of the MRI was performed and the following MR sequences were obtained and compared with the results from the first part: SWI for visualization of the final distribution of SPIO-labeled MSCs, and DWI to exclude the possible embolic events. At the end of the second part of the MR-examination the rats were transferred back to the operating room and after removing the catheter and closing of the operating wound were euthanized as described above without regaining consciousness (approximately 1 h after the start of the IA transplantation). The transcardial perfusion was performed and the brains were removed from the skull to perform histological examination. The study design is schematically presented in Figure 1.

### 2.4. Transient Middle Cerebral Artery Occlusion Model

The 90 min tMCAO modeling was performed as previously described [24]. Briefly, under inhalation anesthesia with isoflurane and mixture of atmospheric air (as described above) in combination with subcutaneous injection of 0.1 mL of 0.5% bupivacaine into the surgery field and intraperitoneal premedication with atropine sulfate 0.05 mg/kg in 1 mL 0.9% NaCl, the bifurcation of the right common carotid artery (CCA) was exposed without affecting the vagus nerve. Following the tMCAO surgical protocol, silicon coated 4-0 monofilament (diameter 0.19 mm, length 30 mm; diameter with coating 0.37 ± 0.02 mm; coating length 3–4 mm, Doccol Corporation, Sharon, MA, USA) was advanced from the right external carotid artery (ECA) to the right internal carotid artery (ICA) until reaching the origin of the middle cerebral artery (MCA) where the slight resistance was felt. For the time of MCA occlusion, the surgical wound was sutured, and the animal was awakened from anesthesia. Ten minutes before the end of the 90 min occlusion period rats were re-anaesthetized, the incision reopened, and the filament was slowly withdrawn providing reperfusion. The operating wound was closed, 3 mL of sterile saline was injected intraperitoneally, and 30 mg/kg gentamicin sulfate was given intramuscularly. The operated rat was placed in a preheated cage for recovery from anesthesia. Rats with hemorrhagic complications were excluded from the experiment (*n* = 1).

### 2.5. Cell Transplantation

Intra-arterial transplantation of MSCs was performed as described previously [7]. Briefly, under inhalation anesthesia with isoflurane/oxygen mix (as described above) the bifurcation of the right CCA was exposed. The pterygopalatine artery was ligated by a 5 ± 0 silk suture, microsurgical clips were placed on the CCA and the ICA, and silk sutures were placed on the ECA. For cell administration microcatheter (MRI applicable rodent tail vein catheter with 1F diameter and 90 cm length, Braintree Scientific, Inc., Braintree, MA, USA) filled with saline was placed into the ECA or the stump of the ECA in case of rats after MCAO, and advanced into the CCA for 5–6 mm in the direction opposite to the blood flow. Our preliminary experiments demonstrated that the reduction of cells’ viability after passing through the catheter was insignificant and that the catheter external diameter is small enough to allow blood flow around it during transplantation [7]. Microsurgical clips were removed to provide maintenance of the blood flow during transplantation, and the rats were put into the MRI scanner located in an adjacent room. After the initial MR examination (see above) and subsequent saline infusion, the catheter was connected to a syringe filled with MSCs suspension and placed in the nanoinjector (Leica Microsystems). IA cell transplantation was performed over a 20 min period. After infusion and the second part of the MR examination, the rats were returned to the operating room, the catheter removed, the ECA stump coagulated and ligated, and the surgical wound was closed.

### 2.6. MRI

All MR-examinations were performed using the 7T ClinScan system for small animals (Bruker BioSpin, Billerica, MA, USA) under inhalation anesthesia with isoflurane as described above. During cell transplantation, isoflurane was mixed with pure oxygen to decrease the concentration of deoxyhemoglobin in the blood and thus reduce the signal from the veins on SWI and improve the visualization of SPIO-labeled MSCs.

After IA catheter insertion, but before the start of IA injection the following sequences were obtained: isotropic T2-weighted images (SPACE, TR\TE  =  4000\251 ms, voxel size 0.2  ×  0.2  ×  0.2 mm, with respiratory trigger), isotropic SWI (susceptibility weighted images, TR\TE  =  40\20 ms, voxel size 0.15  ×  0.13  ×  0.14 mm) with raw data reconstruction (magnitude and phase images), DWI (diffusion-weighted image with calculation of apparent diffusion coefficient maps–ADC, TR/TE  =  15,000/30 ms, b factors  =  0 and 1000 s/mm^2^, voxel size 0.52  ×  0.33  ×  0.5 mm). Selective PWI was performed (EPI, TR\TE  =  1500\15 ms, voxel size 0.3  ×  0.3  ×  1 mm, 220 dynamic series with a 1.5-s duration each one) with intra-arterial Gadolinium contrast injection (Gadobutrol 1 mmol/mL, 0.1 mL/kg). After the end of MSCs transplantation, the same as above isotropic SWI and DWI were obtained.

### 2.7. Data Analysis

The pipeline of data analysis is demonstrated in Figure 2. The script for the main analysis was written using Python 3.8 [25] in Jupyter Notebook [26]; the source code can be found at GitHub (https://github.com/Gubskiy-Ilya/iaPWI_cells access date 20 November 2021).

For the computation of the distribution of transplanted cells in various brain regions, the magnitude images from SWI pulse sequence ensuring better semi-automatic segmentation and visualization in all projections were used.

Processing of the MR data was carried out in 14 steps presented below.

All data were sorted and PWI maps were obtained using perfusion mismatch analyzer (PMA version 5.0.5358.55864, http://asist.umin.jp/ access date 20 November 2021) software (copyright owner: Kohsuke Kudo) provided by ASIST-JAPAN. Maps presenting the following basic parameters were calculated before deconvolution (Figure 3):Time to peak (TTP)-is the time to reach a peak of contrast bolus;Normalized signal drop (ΔS/S or dSoverS)-the relative difference between pre-contrast signal intensity and the lowest signal intensity during the passage of contrast bolus;Maximum slope (MS)-the maximum ratio of signal intensity changes to time;Cerebral blood volume (CBV-AUC)-area under the curve;Additional maps presenting the following basic parameters were calculated after deconvolution analysis based on standard singular value decomposition (sSVD):Cerebral blood flow (CBF-sSVD)-the volume of blood passing through unit volume of brain tissue in one minute;Cerebral blood volume (CBV-sSVD)-the volume of blood in brain tissue;Mean transit time (MTT-sSVD)-length of time during which a certain volume of blood is spent in the cerebral capillary circulation;TTP of residue function (Tmax-sSVD)-the time to a maximum of the residue function obtained by deconvolution.All data were converted to NIfTI file format.The PWI raw data were registered to the first SWI (acquired before the cells’ injection) and the transformation matrix was saved.All PWI maps were registered to the first SWI using calculated transformation matrix.The second SWI acquired after cells’ injection was registered to the first SWI.The cell distribution mask was calculated by finding differences between the first and second SWI.Whole brain segmentation was performed using the U-Net skull stripping tool [27].Obtained brain mask was resampled to the first SWI.The brain mask was cleaned with 3D Slicer (https://www.slicer.org/ access date 20 November 2021 [28]).The cell distribution mask was cropped by the brain mask.The cell distribution mask was finally cleaned with ITK-SNAP (http://www.itksnap.org/ access date 20 November 2021 [29]).Filtered PWI maps were corrupted by the brain mask.The cell distribution density map, was calculated with convolution and application of a median filter to PWI maps was performed.The brain mask, cell distribution density map and filtered PWI maps were used for statistical analysis.

### 2.8. Statistical Analysis

Statistical analysis was performed using Python 3.8 [25] in Jupyter Notebook [26], Spearman correlation implementation from SciPy (https://scipy.org/ access date 20 November 2021). The ordinary least squares linear regression (OLS) implementation from statsmodels (https://www.statsmodels.org/ access date 20 November 2021) was used. GraphPad Prism (https://www.graphpad.com/ access date 20 November 2021) was used to calculate Mann–Whitney U test with FDR correction and for graphical visualization of the final results. Graphs present mean values with standard deviations. The significance level was set at 0.05 and only significant parameters of Spearman correlation and OLS were included in the analysis.

### 2.9. Immunocytochemistry and Microscopy

Animals were sacrificed 1 h after IA transplantation of MSCs by inhalation anesthesia with a lethal dose of isoflurane and additional injection of a lethal dose of Zoletil. Transcardial perfusion was carried out using phosphate-buffered saline (PBS, 0.1 M, pH 7.4), followed by icecold 4% paraformaldehyde in 0.1 M PBS. After decapitation, the brains were removed from the skull, held at 4 °C overnight in the same fixative, washed three times with PBS, and immersed in 30% sucrose solution. Coronal sections 40 μm thick were obtained using a cryostat microtome (Leica CM1900). Sections containing transplanted cells were selected by the fluorescence of Dragon green SPIO and PKH26 markers. Then sections were mounted on slides and processed for immunohistochemistry. For nonspecific binding blocking, sections were incubated in a blocking solution consisting of PBS containing 5% normal goat serum and 0.1% Triton-X 100 for 30 min at room temperature. The specimens were then incubated at 4 °C overnight with primary anti-Mitochondria antibody (1:100, Abcam, Cambridge, UK) diluted in blocking solution. Then sections were rinsed 3 times for 10 min with PBS. Then specimens were incubated with the secondary antibodies anti-mouse IgG Alexa fluor 647 (1:500, Abcam, Cambridge, UK) for 2 h at room temperature (21–24 °C). For the nuclei counterstaining DAPI solution (2 µg/mL, Sigma-Aldrich, Burlington, MA, USA) was used. Sections were coverslipped with 80% glycerol. Laser scanning confocal microscope Nikon A1R MP + was used for image acquisition.

## 3. Results

### 3.1. Visualization of MSCs Distribution in Rat Brain and Perfusion Maps

In the rat brain, SPIO-labeled MSCs after transplantation can be visualized as hypointense spots on SWI, since the SPIO label reduces T2* relaxation time. This pulse sequence has proven to be the most sensitive for in vivo detection of small groups or merely single SPIO-labeled cells after systemic administration [22]. Transplanted cells were distinguished from cerebral blood vessels by their shape on the series of adjacent slices. Moreover, for improving MSCs’ visualization the mixture of isoflurane with pure oxygen was used for inhalation anesthesia, which allowed to reduce the signal from the veins on SWI due to reduction of deoxyhemoglobin concentration in the blood. Additionally, MRI data of cells distribution were verified by histological examination: double cell labeling (SPIO-dragon green and PKH26) combined with immunohistochemical staining with antibodies against human mitochondria allowed detection of human MSCs in the rat brain sections and its subsequent comparison with MR-images. In both intact rats and animals with experimental stroke, MSCs after IA injection into the right ICA were distributed predominantly in the right brain hemisphere and only a few of them entered the contralateral (left) hemisphere. In this study, in accordance with our previous results [7], the majority of labeled MSCs were found inside cerebral blood vessels of the cerebral cortex and subcortical regions (thalamus, striatum, hypothalamus, and hippocampus), corpus callosum, and the brain stem. Examples of obtained MR and histological images are presented in Figure 4. It is important to note that in all rats included in the study no thromboembolic complications were detected after IA MSCs transplantation confirming the safety of our cell infusion parameters. The infusion parameters used in this study were selected with regard to literature data [6] and on the basis of our previous works [7,23]. In the current study, we also controlled the safety of transplantation for each rat by performing DWI with apparent diffusion coefficient maps (ADC) calculation before and after IA administration of MSCs. This pulse sequence allows for visualization of the zones of cytotoxic edema (regions with restricted diffusion of free water molecules) and is widely used for early diagnostics of cerebral ischemia in clinical practice and basic research [30,31,32]. Additional histological evaluation of brain sections after IA injection of MSCs for evaluation of the possibility of neuronal death in the zones of cells’ accumulation can be found in the Appendix A.

For all rats included in the experiment, a selective IA MR-perfusion study was performed just before IA administration of MSCs, and different PWI maps were obtained. During visual assessment and comparison of the perfusion maps and SWI, we observed a mild degree of overlap between the location of the areas with high levels of cerebral perfusion and areas of SPIO-labeled cells accumulation for both intact rats and rats with tMCAO. An example of similarity of cerebral blood volume map (CBV-AUC) and hypointense spots on SWI is demonstrated in Figure 5. Similar data were obtained for other parametric maps (TTP, dSoverS, MS, CBF-sSVD, CBV-sSVD, MTT-sSVD, Tmax-sSVD) and then the quantitative analysis of correlations between different perfusion parameters and the distribution of MSCs was performed (see the next section).

### 3.2. Evaluation of the Difference between the Two Experimental Groups

At first, we separately measured and then compared different quantitative perfusion parameters (TTP, MS, dSoverS, Tmax-sSVD, MTT-sSVD, CBV-AUC CBV-sSVD, CBF-sSVD) with final cell density measured immediately after the end of cell transplantation in the whole brain for rats after tMCAO and for intact animals. For estimation of correlation, we used the Spearman correlation coefficient. The adjusted R-Squared coefficient of determination from ordinary least squares linear regression was used to estimate the variability of cell density that can be predicted from perfusion maps. After comparison of these parameters for both experimental groups no statistically significant differences were found (data are given in Figure 6). Based on the obtained data, we pooled both groups for further analysis.

### 3.3. Correlation between Brain Perfusion Parameters and Cell Density in the Whole Brain

At the second stage, we compared different quantitative perfusion parameters with final cell density in the whole brain for all experimental animals. We cropped the perfusion maps using a full brain mask to exclude the extracranial structures from the analysis. We used the Spearman correlation coefficient for estimation of the correlation and coefficient of determination to estimate the variability of cell density that can be predicted from perfusion maps. We obtained moderate (0.53–0.55) Spearman correlation between final cell density and dSoverS, CBV-AUC, CBV-sSVD, CBF-sSVD parametric maps (data presented in Figure 7A). The maximum coefficients of determination (23–25%) were calculated for CBF-sSVD, CBV-sSVD, CBV-AUC, and dSoverS perfusion parameters (data shown in Figure 7B). For timing parameters, TTP and MTT-sSVD, the correlation was very weak. For Tmax-sSVD parameter the correlation was a moderate negative, but with the coefficient of determination around zero.

### 3.4. Correlation between Brain Perfusion Parameters and Cell Density in the Areas of Their Accumulation

The estimate of the Spearman correlation coefficient between perfusion parameters and cell density in the whole brain could be altered since the contrast agent and the cells were injected unilaterally via the right ICA. Consequently, they were distributed predominantly within the right hemisphere, while the brain mask included zones outside the area of blood supply territory of the right ICA. Thereby, we cropped all perfusion maps by removing regions without transplanted cells and repeated the analysis to compare quantitative perfusion parameters with final cell density in the place of their accumulation for all experimental animals. For all timing parameters such as TTP, Tmax-sSVD, and MTT-sSVD the correlation was very weak. We obtained weak correlation (0.31–0.33) for CBF-sSVD, CBV-sSVD, CBV-AUC, and dSoverS perfusion parameters (data presented in Figure 7C). The maximum coefficients of determination (28–30%) were calculated for CBF-sSVD, CBV-sSVD, CBV-AUC, and dSoverS perfusion parameters (data shown in Figure 7D).

## 4. Discussion

In this study, we saw the momentary arrangement of transplanted MSCs in the rat brain, registered by SWI immediately after 20 min long infusion of cell suspension through the ICA, and reflecting the net result of the entrapment of cells in cerebral vessels and their return to circulation. Histological images (Figure 5) clearly show that during the time of infusion embracing many passages of blood, MSCs remained within the vessels and did not transmigrate into the brain parenchyma. Cells were really trapped within the vessels, since otherwise they would probably be washed out during the procedure of transcardial perfusion employed in the histological study, and since the DWI check performed immediately after SWI excluded the thrombosis of the capillaries with cell aggregates. The further fate of the entrapped cells can be different, including temporary entrapment, return to circulation, and crossing the blood-brain barrier followed by invasion into the brain parenchyma (reviewed in [11]). However, in any case, the initial distribution of transplanted cells is important.

Before cell administration, we used selective MR-perfusion with IA injection of gadolinium-based contrast agent via the ICA as a tool to evaluate the cerebral perfusion impact on the distribution of MSCs after IA transplantation in rat brain of intact animals and after experimental stroke modeling. The transcatheter intra-arterial perfusion, also designated as selective perfusion study with bolus contrast agent injection allows estimation of the perfusion of a chosen vessel and its blood supply territory with arterial blood [33,34,35]. This technique has been used for a long time in clinical oncology to predict the distribution of chemoembolic materials during trans-arterial chemoembolization [34,35], and can be adjusted to cell therapy applications in clinical setup. One of the limitations of gadolinium for cerebral perfusion evaluation is the effect of contrast leakage during the first pass under different pathologic conditions characterized by the disruption of the blood brain barrier (BBB) [36]. T2*-contrast agents, including SPIO particles used by Walczak et al. [18], can also be used for MR-perfusion study. The advantage of SPIO particles is their larger size preventing them from passing through the blood brain barrier. On the other hand, gadolinium-based contrast agents are commonly utilized in clinical practice and in some countries are the only ones allowed for clinical applications [37].

In our experiments, in accordance with data reported by Walczak et al. [18], visually there was a certain degree of overlap between areas with high cerebral perfusion and MSCs accumulation assessed by SWI. We used two approaches to data analysis: the first employs whole brain masks, and the second-cell distribution masks. In our opinion, a more accurate estimate of the impact of cerebral perfusion was provided by the second method, since it did not include zones outside the area of blood supply territory of the right ICA. The results obtained by this method show that the distribution of IA transplanted MSCs does correlate with brain perfusion in both intact rats and rats with experimental stroke. However, in our hands, the coefficient of determination did not exceed 30%. This means that based on the perfusion data, the density of cell distribution in the brain after IA transplantation is determined by the perfusion for no more than 30%. The method using the whole brain masks has slightly lower reliability (23–25%), but, on the other hand, it allows to roughly predict the post-injection stem cell distribution. Potentially, this may help in clinical decision-making when choosing the IA positioning of the catheter and correcting the transplantation parameters (cell dose, infusion volume, velocity and etc.) based on individual perfusion characteristics of the patients.

Our results have demonstrated that it is not necessary to use complex methods of perfusion data processing for prediction of cell distribution, such as deconvolution. It is enough to evaluate the dynamics of signal changes during the passage of contrast agent and calculate quantitative basemaps, such as dSoverS and CBV-AUC. The CBF-sSVD and CBV-sSVD obtained after deconvolution had similar or slightly lower correlation values with the final distribution of cells. Time maps TTP, MTT-sSVD, and Tmax-sSVD did not show any significant prognostic value. Interestingly, while comparing different quantitative perfusion parameters with final cell density we found no significant differences between rats after tMCAO and intact animals. The number of animals in the intact group was smaller than in tMCAO group, which we would mention as a limitation of this study. However, others have also demonstrated similar distribution of MSCs after intra-arterial administration in both healthy animals and animals with modeled brain infarction [38,39]. The reasons for this similarity are currently unknown and require detailed studies with consideration of various factors.

The results of this study demonstrate that mesenchymal stem distribution after intra-arterial transplantation cannot be explained exclusively by the perfusion data. Other factors, such as adhesion properties of MSCs [40] and of the vascular wall determined by the availability and affinity of cell adhesion molecules expressed by endothelium and transplanted cells, the BBB integrity, local chemokine concentrations, etc., may also affect cell distribution [8,41,42]. We also cannot rule out the possibility of transplanted cells entrapment inside small vessels located at the border of the blood supply territories of brain arteries (watershed zones) [43,44]. Further research is needed to fully characterize the factors contributing to IA transplanted MSCs distribution.

In addition, the proposed method of quantitative assessment of the impact of cerebral perfusion on mesenchymal stem cells distribution after intra-arterial transplantation can be also used for other cell types in future studies.

## Figures and Tables

**Figure 1 biomedicines-10-00353-f001:**
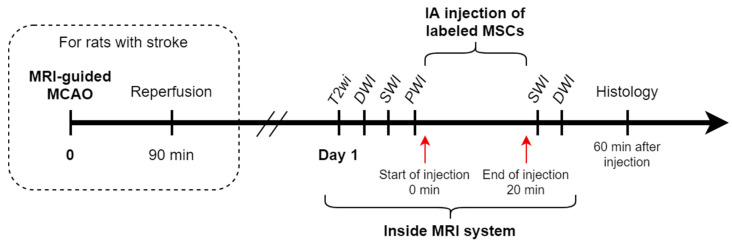
Study design. The steps of the study are schematically presented in a timeline.

**Figure 2 biomedicines-10-00353-f002:**
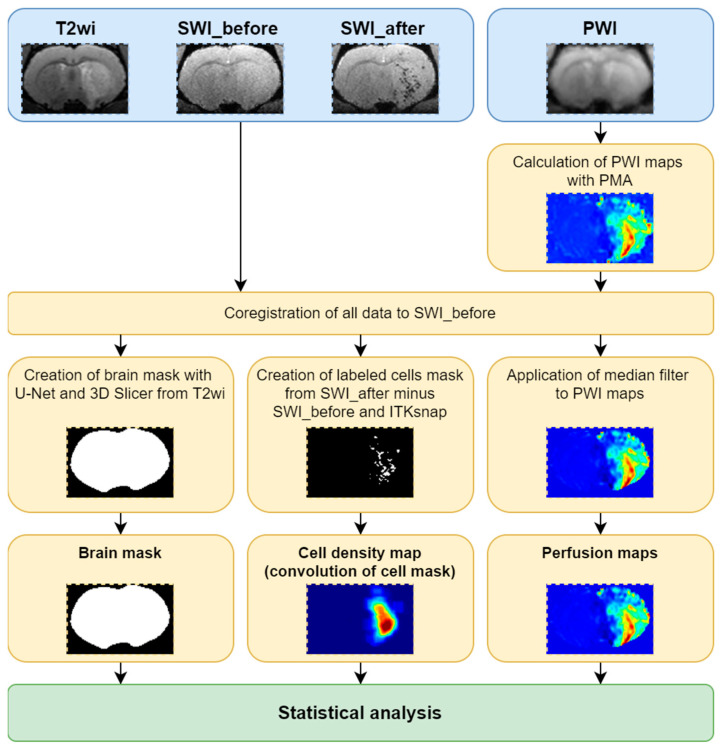
The summary of the pipeline of data analysis: schematic representation of the steps of analysis are schematically presented.

**Figure 3 biomedicines-10-00353-f003:**
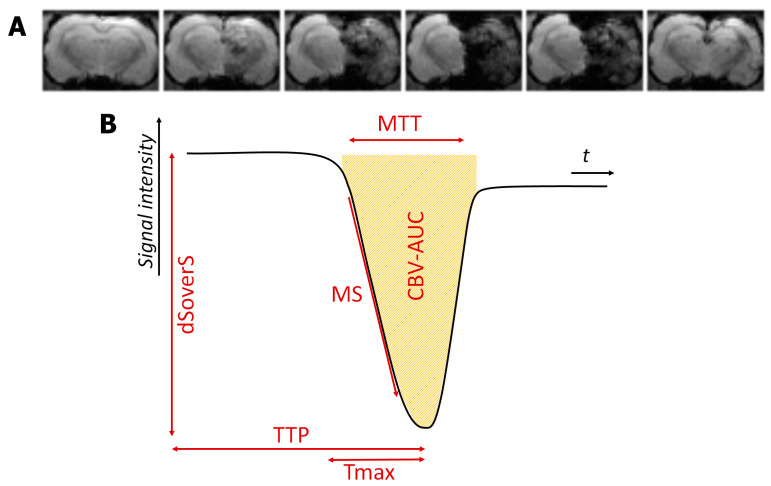
The example of dynamic T2*-weighted images of the perfusion study during selective intra-arterial contrast injection (**A**) and resulting perfusion curve with different quantitative parameters (**B**).

**Figure 4 biomedicines-10-00353-f004:**
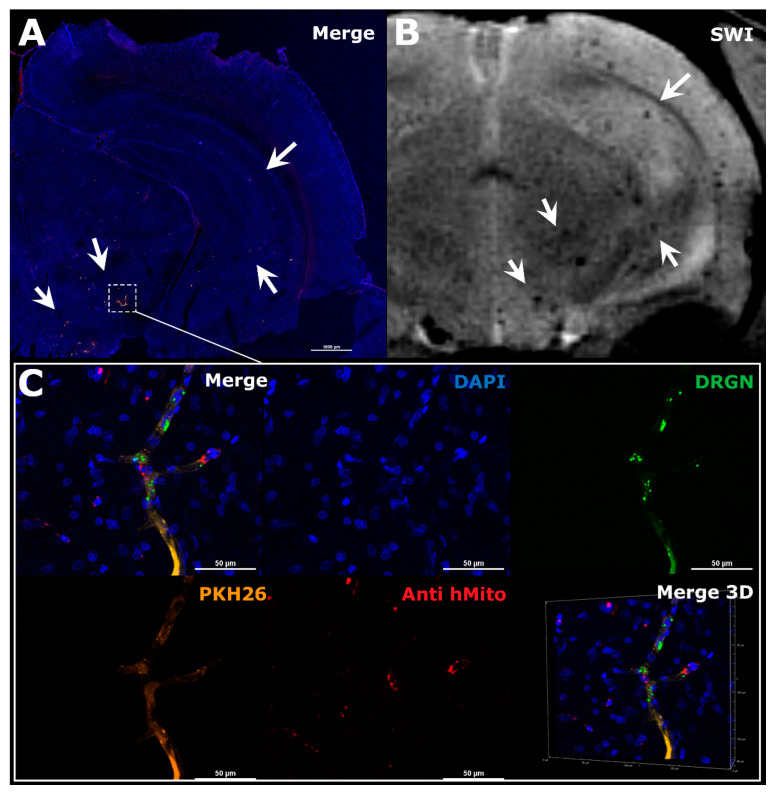
The comparison of MRI and histological images of the rat brain after IA administration of MSCs. The example illustrated the distribution of MSCs in the right hemisphere of the ischemic rat brain during the first hour after transplantation. (**A**) Panoramic confocal fluorescence image of the coronal sections of rat brain 1 h after MSCs’ transplantation. White rectangle and arrows indicate zones of labeled cell location. Before transplantation human MSCs were double-labeled with membrane lipophilic dye PKH26 (orange) and SPIO microparticles conjugated with fluorescent marker Dragon Green with cytoplasmic accumulation (green). Additionally, transplanted cells were stained using antibodies against human mitochondria (red). The nuclei were stained with DAPI (blue). Scale bar: 1000 μm. (**B**) SWI of the same rat brain 30 min after IA injection, white arrows indicated hypointense spots of SPIO-labeled cells accumulation. MRI data on cell distribution coincide with the histological study. (**C**) High-magnification confocal fluorescent images of the rat brain from the zone marked by the rectangle in A. Transplanted double-labeled MSCs were located inside cerebral blood vessels 1 h after IA transplantation (PKH26–orange, SPIO microparticles with fluorescent marker Dragon Green–green, MSCs stained using antibodies against human mitochondria–red, nuclei stained with DAPI–blue). Scale bars: 50 μm. Bottom right: 3D-reconstruction of z-stacks.

**Figure 5 biomedicines-10-00353-f005:**
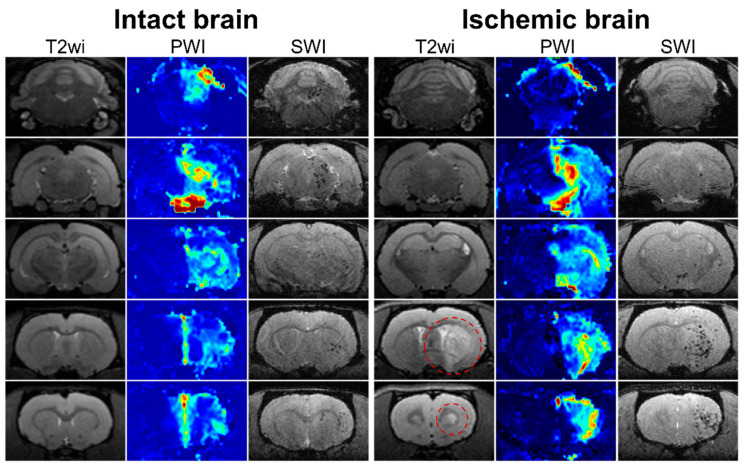
MRI data of rat brain of intact animal (left column) and rat with experimental stroke (right column). In the columns five sequential coronal brain sections visualized using different pulse sequences are presented: T2 weighted images before MSCs transplantation for visualization of brain anatomy and ischemic lesion (the hyperintense zones marked with red circle); perfusion map (CBV-AUC) obtained after injection of gadolinium-based contrast agent through the right ICA before cell administration; SWI 20 min after the start of IA transplantation of labeled MSCs through the right ICA for visualization of transplanted cells distribution. SPIO labeled cells are hypointense (dark) spots on SWI. In both intact and ischemic rat brains mild similarity between the location of the areas with high levels of cerebral perfusion and areas of SPIO-labeled cells accumulation is observed.

**Figure 6 biomedicines-10-00353-f006:**
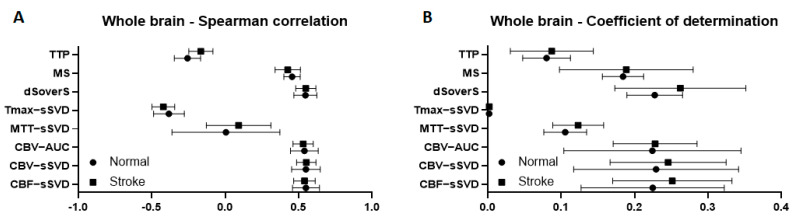
Correlation between different brain perfusion parameters and cell density in a whole brain for rats with stroke and intact animals. (**A**) graphs of Spearman correlation coefficients between perfusion maps and cell density; (**B**) graphs of coefficient of determination show the variability of the cell density that can be predicted from the perfusion maps. Graphs present mean values with standard deviations.

**Figure 7 biomedicines-10-00353-f007:**
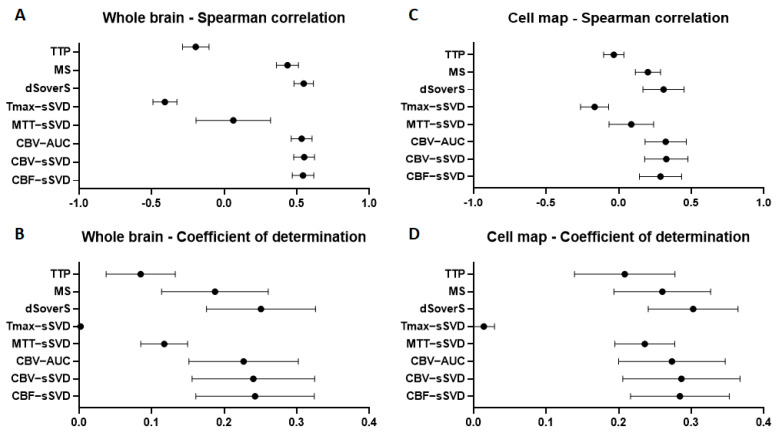
Correlation between different brain perfusion parameters and cell density in the whole brain (**A**,**B**) and in the places of their accumulation (**C**,**D**). (**A**,**C**)—graphs of the Spearman correlation coefficient between perfusion maps and cell density; (**B**,**D**)—graphs of the coefficient of determination show the proportion of the cell density that can be predicted from the perfusion maps.

## Data Availability

The data presented and analyzed in this study can be available from the corresponding author upon reasonable request.

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
