# Peer review of "The Impact of Cerebral Perfusion on Mesenchymal Stem Cells Distribution after Intra-Arterial Transplantation: A Quantitative MR Study"

_biomedicines, 2022, doi:10.3390/biomedicines10020353_

Round 1
Reviewer 1 Report
The authors wanted to present a study regarding the impact of cerebral perfusion on MSC distribution post IA infusion.
The design was nicely presented and performed. The experiment is well controlled in general.
Question to authors 1 : Did you try the same model with MSC from other sources of MSC ?
Question to authors 2 : Did you try to use as control another type of cells as fibroblasts ?
Author Response
We would like to thank you for reviewing our manuscript and your expert opinion. Please, find the answers to your questions below.
- «Question to authors 1: Did you try the same model with MSC from other sources of MSC?».
Over the past 5 years, since 2016 our team concentrated on studying the effects of the transplantation of mesenchymal stem cells derived from human placenta in the rat MCAO experimental ischemic stroke model and deciphering the underlying cellular mechanisms. The results are presented in the following publications: doi: 10.1371/journal.pone.0186717; doi: 10.1088/1742-6596/886/1/012012; doi: 10.3389/fnins.2021.641970; doi: 10.1007/s12975-017-0590-y; doi: 10.3390/cells10112997. This particular cell type was chosen, because the totality of the placenta-derived MSC features makes this type of cells one of the best candidates for the allogenic transplantation in acute stroke patients. We also tried to minimize the number of animals used in the experiments in order to meet both the international and domestic animal use regulations. Accordingly, working with placenta-derived MSCs, we did not attempt to transplant the MSCs isolated from other sources.
- «Question to authors 2: Did you try to use as control another type of cells as fibroblasts?».
At the planning stage of our experiment, we successfully transplanted neural progenitor cells obtained by direct reprogramming (drNPC). However, we decided to focus on MSCs for the reasons described above and to avoid overloading the submitted paper. Preliminary experimental data for intra-arterial transplantation of drNPC (qualitative analysis of cerebral perfusion and cell distribution) are presented below on Figure 1.
https://drive.google.com/file/d/11l7M70oFQfjR7WRKfqZVMqPxDN9u1HnW/view?usp=sharing
Figure 1. The figure presents MRI data of rat brain after IA transplantation of labeled drNPCs through the right internal carotid artery 24 h after the modeling of experimental ischemic stroke in the right middle cerebral artery territory. In the columns 4 sequential coronal brain sections, visualized using different pulse sequences, are presented: T2 weighted images before cells transplantation for visualization of brain anatomy and ischemic lesion (the hyperintense zones marked with red dotted line); SWI before and after of IA transplantation of labelled cells through the right ICA (transplanted cells marked with the red arrow); perfusion map (CBV-AUC) obtained after injection of gadolinium-based contrast agent through the right ICA before cell administration. SPIO labeled cells are hypointense (dark) spots on SWI. The examination was performed with air (not with pure oxygen), so cerebral veins are visualized as hypointense linear and dotted structures interfering with the quality of cells distribution assessment. Moderate level of similarity between the location of the areas with high levels of cerebral perfusion and areas of SPIO-labelled cells accumulation is observed.
The similarity between cerebral perfusion and distribution of cells from different sources has been well described by Walczak et al. in the following paper: doi: 10.1177/0271678X16665853. We believe, that the method of quantitative assessment of the impact of cerebral perfusion on mesenchymal stem cells distribution after intra-arterial transplantation proposed in our study can be used for the other cell types in future studies. We added this to the discussion section.
Best regards,
Ilya Gubskiy, MD, PhD
Konstantin Yarygin, MD, PhD
Reviewer 2 Report
Manuscript entitled “The impact of cerebral perfusion on mesenchymal stem cells distribution after intra-arterial transplantation: A quantitative MR study” studied the distribution of MSCs in the brain after injection into the internal carotid artery (ICA). The study provided an efficient method to determine the distribution of MSCs after ICA injection. The results showed that MSCs were entrapped in the small blood vessels assumed to be arterioles in intact rats and rats with MCAO 60 min after injection, confirmed previous findings in other organs. The study provides interesting data to understand the distribution of MSCs after ICA injection, a routes to deliver drugs to diseased brain particularly in the case of ischemic strokes. However, there are several issues to clarify to support some statements in the manuscript.
- The study showed that ICA injection of MSCs into intact rats and rats with MCAO yielded similar results in the distribution of MSCs, suggesting that MSCs distribution in the brain was mainly determined by their mechanical properties, and that the interactions of the inflamed endothelium with MSCs were barely involved. While this might be the case, the sample size of the intact rat group should be increased to 7 to reduce impact of variations among animals.
- MSCs used in this study should be characterized carefully, including the cell size and surface expression of adhesion molecules (receptors), particularly those potentially mediating cell adhesion to endothelial cells.
- The study attempted to state that the entrapment of MSCs in the terminal arteries (assumed by images in Figure 4, where the lumen of the blood vessel with entrapped MSCs was about 10-20 um in diameter) did not cause thrombosis. As this is quite unlikely as the blood vessel lumen was so small that could easily be occluded by a few MSCs, and this is against previous findings, where intra-artery delivery of MSCs in small and large animals (dog and pig) caused microinfarction. Therefore, I suggest that the authors to examine brain sections of rats after ICA injection of MSCs for neuronal death in the perfused region.
Author Response
Thank you very much for reviewing our manuscript. We highly appreciate your comments and tried our best to answer your questions and make the necessary amendments according to your suggestions.
- «The study showed that ICA injection of MSCs into intact rats and rats with MCAO yielded similar results in the distribution of MSCs, suggesting that MSCs distribution in the brain was mainly determined by their mechanical properties, and that the interactions of the inflamed endothelium with MSCs were barely involved. While this might be the case, the sample size of the intact rat group should be increased to 7 to reduce impact of variations among animals».
The main goal of this study was quantitative assessment of the impact of brain tissue perfusion estimated by MRI on the distribution of IA transplanted MSCs in intact rats and rats with experimental stroke, which we tried to underline in the introduction section. Our results demonstrated certain correlation between the distribution of IA transplanted MSCs and brain perfusion with the coefficient of determination up to 30% in both intact rats and rats with experimental stroke. Interestingly, others have also demonstrated similar distribution of MSCs after intra-arterial administration in both healthy animals and animals with modeled brain infarction [doi: 10.3727/096368914X679336; doi: 10.1016/j.expneurol.2012.09.018]. The reasons for this similarity are currently unknown and require detailed studies with consideration of various factors. We speculate about this in the Discussion section. Besides cerebral perfusion, we listed other factors, that may have impact on MSC distribution, including besides cerebral perfusion cells’s mechanical properties and their interactions with endothelium, as you correctly mentioned.
According to the table of critical values for the nonparametric Mann-Whitney U Test and two-sided level of significance (α=0.05), it can be used when analyzing groups containing 3 and 7 values. The power of such analysis may not be high, but if there are pronounced differences, they manifest themselves. The main statistical analysis with Spearman correlation and ordinary least squares linear regression was carried out with samples of 200 000 or more values in each group for assessment of correlation between brain perfusion parameters and cell density in the areas of their accumulation and 700 000 or more values in the case of assessment correlation between brain perfusion parameters and cell density in the whole brain. However, in the text we mentioned the number of animals in the group of intact animals as a limitation of our study.
- MSCs used in this study should be characterized carefully, including the cell size and surface expression of adhesion molecules (receptors), particularly those potentially mediating cell adhesion to endothelial cells.
The adhesion properties of MSCs and vascular wall undoubtedly play an important role in the distribution of MSCs after transplantation. We stress it in the Discussion section. However, the main goal of the current study was quantitative assessment of the impact of brain tissue perfusion estimated by MRI on the distribution of IA transplanted MSCs in intact rats and rats with experimental stroke. We did not investigate the other possible factors, that can have also impact on cells distribution.
- The study attempted to state that the entrapment of MSCs in the terminal arteries (assumed by images in Figure 4, where the lumen of the blood vessel with entrapped MSCs was about 10-20 um in diameter) did not cause thrombosis. As this is quite unlikely as the blood vessel lumen was so small that could easily be occluded by a few MSCs, and this is against previous findings, where intra-artery delivery of MSCs in small and large animals (dog and pig) caused microinfarction. Therefore, I suggest that the authors to examine brain sections of rats after ICA injection of MSCs for neuronal death in the perfused region.
Intra-arterial transplantation is an affective delivery route for stem cells administration, however, it can cause additional thromboembolic strokes in case of incorrect transplantation parameters used. Guzman et al. [doi: 10.1161/strokeaha.117.018288] analyzed this problem in detail in an excellent review in «Stroke» journal and identified several factors crucial for the transplantation outcome: cell type and size, cell dose, infusion speed, treatment window, and the extent of preservation of the natural arterial blood flow in the feeding vessel. With the use of selected for the object and cell type parameters the IA delivery can be safe. This is also confirmed by the results of the first clinical trials dedicated to the safety of IA MSCs transplantation in stroke patients [reviewed in doi: 10.3389/fcell.2021.621645]. The infusion parameters used in this study were selected with regard to the literature data and based on our previous works [doi: 10.1088/1742-6596/886/1/012012, doi: 10.3389/fnins.2021.641970], where we investigated and discussed in detail the safety of transplantation. In the current study, despite using the selected close-to-optimum parameters, we controlled the safety of transplantation for each rat by performing the diffusion weighted MR imaging (DWI) with apparent diffusion coefficient maps (ADC) calculation before and after IA administration of MSCs. This pulse sequence allows to visualize zones with the cytotoxic edema (regions with restricted diffusion of free water molecules) and is widely used in clinical practice and basic research for early diagnostics of cerebral ischemia [doi: 10.1148/radiology.210.1.r99ja02155; PMCID: PMC8367476; doi: 10.1161/STR.0000000000000375]. The voxel size of DWI pulse sequence used in our study was 0.33x0.33x0.5 mm, which means that detection threshold for visualization of cerebral infarction was approximately 55 nanoliters. We tried to write this more clearly in the results section of the amended paper. Additionally, we performed histological analysis of brain sections of rats after IA injection of MSCs to evaluate the possibility of neuronal death in the perfused regions and zones of cells’ accumulation. To do this we performed TUNEL analysis for detection of apoptotic cells. Our results verify the MRI data and confirm that there were no apoptotic brain cells in the zones of MSCs accumulation after transplantation (except for the infarction area). The obtained results can be found in the supplementary material and below.
https://drive.google.com/file/d/19LZI_u53SipTRlR5aX0g1_IykXAl6DSX/view?usp=sharing
Figure 1. The safety of MSCs IA transplantation. The comparison of MRI and histological images of the brain of rat with experimental stroke after IA administration of labeled MSCs confirmed the absence of new infarction lesions formation. (A) T2 weighted images and Apparent Diffusion Coefficient (ADC) map before cells transplantation. The red lines indicate the ischemic lesion after MCAO stroke modeling, which is hyperintense on T2wi with restricted diffusion on ADC. (B) Apparent Diffusion Coefficient (ADC) map and SWI after cells MSCs IA transplantation. White arrows indicate labeled cells (hypointense dots on SWI). In the places of MSCs accumulation no additional lesions with restricted diffusion were detected (blue line area on ADC). (C) High-magnification fluorescent images of the rat brain from the zone with labeled cells accumulation outside the ischemic lesion (marked by the green rectangle in B). Transplanted labeled MSCs marked with white arrows, SPIO microparticles with fluorescent marker Dragon Green - green, nuclei stained with DAPI - blue. TUNEL+ signal (red) of apoptotic cells is negative. Scale bars: 50 μm. (D) High-magnification fluorescent images of the rat brain in the ischemic lesion (marked by the red rectangle in B). TUNEL+ signal (red) of apoptotic cells marked with white arrows, nuclei are also stained with DAPI (blue). Scale bars: 50 μm.
Best regards,
Ilya Gubskiy, MD, PhD
Konstantin Yarygin, MD, PhD

Round 2
Reviewer 2 Report
The experiment sample size should be larger. Additional experiments are needed to support your result.